# Differentially Private Diffusion Models Generate Useful Synthetic Images

## Abstract

The ability to generate privacy-preserving synthetic versions of sensitive image datasets could unlock numerous ML applications currently constrained by data availability. Due to their astonishing image generation quality, diffusion models are a prime candidate for generating high-quality synthetic data. However, recent studies have found that, by default, the outputs of some diffusion models do not preserve training data privacy. By privately fine-tuning ImageNet pre-trained diffusion models with more than 80M parameters, we obtain SOTA results on CIFAR-10 and Camelyon17 in terms of both FID and the accuracy of downstream classifiers trained on synthetic data. We decrease the SOTA FID on CIFAR-10 from 26.8 to 9.8, and increase the accuracy from 51.0% to 88.0%. On synthetic data from Camelyon17, we achieve a downstream accuracy of 91.1% which is close to the SOTA of 96.5% when training on the real data. We leverage the ability of generative models to create infinite amounts of data to maximise the downstream prediction performance, and further show how to use synthetic data for hyperparameter tuning. Our results demonstrate that diffusion models fine-tuned with differential privacy can produce useful and provably private synthetic data, even in applications with significant distribution shift between the pre-training and fine-tuning distributions.

## 1 Introduction

Delivering impactful ML-based solutions for real-world applications in domains like health care and recommendation systems requires access to sensitive personal data that cannot be readily used or shared without risk of introducing ethical and legal implications. Replacing real sensitive data with private synthetic data following the same distribution is a clear pathway to mitigating these concerns (Patki et al., 2016; Dankar & Ibrahim, 2021; Chen et al., 2021b). However, despite their theoretical appeal, general-purpose methods for generating useful and provably private synthetic data remain a subject of active research (Dockhorn et al., 2022; McKenna et al., 2022; Torfi et al., 2022). The central challenge in this line of work is how to obtain truly privacy-preserving synthetic data free of the common pitfalls faced by classical anonymization approaches (Stadler et al., 2021), while at the same time ensuring the resulting datasets remain useful for a wide variety of downstream tasks, including statistical and exploratory analysis as well as machine learning model selection, training and testing.

It is tempting to obtain synthetic data by training and then sampling from well-known generative models like variational auto-encoders (Kingma & Welling, 2013; Rezende et al., 2014), generative adversarial nets (Goodfellow et al., 2020), and denoising diffusion probabilistic models (Song & Ermon, 2019; Ho et al., 2020). Unfortunately, it is well-known that out-of-the-box generative models can potentially memorise and regenerate their training data points[1] and, thus, reveal private information. This holds for variational autoencoders (Hilprecht et al., 2019), generative adversarial nets (Hayes et al., 2017), and also diffusion models (Carlini et al., 2023; Somepalli et al., 2022; Hu & Pang, 2023; Duan et al., 2023). In particular, diffusion models have recently gained a lot of attention, with pre-trained models made available online

---

[1]A model does not contain its training data, but rather has "memorised" training data when the model is able to use the rules and attributes it has learned about the training data to generate elements of that training data.

CIFAR-10                                                 Camelyon17

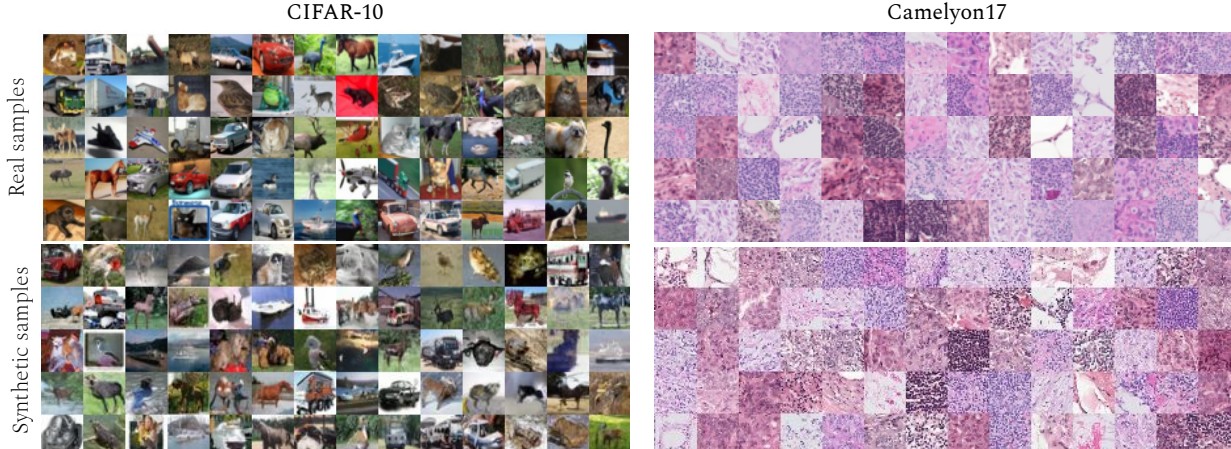

Figure 1: DP diffusion models are capable of producing high-quality images. More images can be found in Figures 5, 6, 7.

(Dhariwal & Nichol, 2021; Rombach et al., 2022), and being fine-tuned on potentially sensitive data such as chest X-rays (Chambon et al., 2022b; Ali et al., 2022; Chambon et al., 2022a) and brain MRIs (Rouzrokh et al., 2022; Pinaya et al., 2022).

Mitigating the privacy loss from sharing synthetic data produced by generative models trained on sensitive data is not straightforward. Differential privacy (DP) (Dwork et al., 2006) has emerged as the gold standard privacy mitigation when training ML models, and its application to generative models would provide guarantees on the information the model (and synthetic data sampled from it) can leak about individual training examples. Yet, scaling up DP training methods to modern large-scale models remains a significant challenge due to the slow down incurred by DP-SGD (the standard workhorse of DP for deep learning) (Wang et al., 2017) and the stark utility degradation often observed when training large models from scratch with DP (Zhang et al., 2022; Stadler et al., 2021; Stadler & Troncoso, 2022). Most previous works on DP generative models worked around these issues by focusing on small models, low-complexity data (Xie et al., 2018; Torkzadehmahani et al., 2019; Harder et al., 2021) or using non-standard models (Harder et al., 2022). However, for DP applications to image classification it is known that using models pre-trained on public data is a method for attaining good utility which is compatible with large-scale models and complex datasets (Bu et al., 2022a; De et al., 2022; Cattan et al., 2022; Xu et al., 2022; Tramèr et al., 2022; Bu et al., 2022b).

**Contributions.** In this paper we demonstrate how to accurately train standard diffusion models with differential privacy. Despite the inherent difficulty of this task, we propose a simple methodology that allows us to generate high-quality synthetic image datasets that are useful for a variety of important downstream tasks. In particular, we privately train denoising diffusion probabilistic models (Ho et al., 2020) with more than 80M parameters on CIFAR-10 and Camelyon17 (Koh et al., 2021), and evaluate the usefulness of synthetic images for downstream model training and evaluation. Crucially, we show that by pre-training on publicly available data (i.e. ImageNet), it is possible to considerably outperform the results of extremely recent work on a similar topic (Dockhorn et al., 2022). With this method, we are able to accurately train models 45x larger than Dockhorn et al. (2022) and to achieve a high utility on datasets that are significantly more challenging (e.g. CIFAR-10 and a medical dataset – instead of MNIST). Please refer to Table 2 for a detailed comparison of our works. Our contributions can be summarized as follows:

- We demonstrate that diffusion models can be trained with differential privacy to sufficient quality that we can create accurate classifiers based on the synthesized data only. To do so, we leverage pre-training, and we demonstrate large state-of-the-art (SOTA) improvements even when there exists a significant distribution shift between the pre-training and the fine-tuning data sets.

- We propose simple and practical improvements over existing methods to further boost the performance of the model. Namely, we employ both image and timestep augmentations when using augmentation multiplicity, and we bias the diffusion timestep sampling so as to encourage learning of the most challenging phase of the diffusion process.

- With this approach, we fine-tune a differentially private diffusion model with more than 80 million parameters on CIFAR-10, and beat the previous SOTA by more than 50%, decreasing the Fréchet Inception Divergence (FID) from 26.8 to 9.8. Furthermore, we privately fine-tune the same model on histopathological scans of lymph node tissue available in the Camelyon17 dataset and show that a classifier trained on synthetic data produced by this model achieves 91.1% accuracy (the highest accuracy reported on the WILDS leaderboard (Koh et al., 2021) is 96.5% for a non-private model trained on real data).

- We demonstrate that the accuracy of downstream classifiers can be further improved to a significant extent by leveraging larger synthetic datasets and ensembling, which comes at no additional privacy cost. Finally, we show that hyperparameter tuning downstream classifiers on the synthetic data reveals trends that are also reflected when tuning on the private data set directly.

## 2 Related Work

**Differentially private synthetic image generation.** DP image generation is an active area of research (Fan, 2020; Croft et al., 2022; Chen et al., 2022b). Most efforts have focused on applying a differentially private stochastic gradient procedure on popular generative models, i.e. generative adversarial networks (Xie et al., 2018; Jordon et al., 2018; Torkzadehmahani et al., 2019; Augenstein et al., 2019; Xu et al., 2019a; Liu et al., 2019; Chen et al., 2020; Yoon et al., 2020; Chen et al., 2021a), or variational autoencoders (Pfitzner & Arnrich, 2022; Jiang et al., 2022). Only one other work has so far analysed the application of differentially private gradient descent on diffusion models (Dockhorn et al., 2022) which we contrast against in Table 2. Others have instead proposed custom architectures (Chen et al., 2022a; Wang et al., 2021; Cao et al., 2021; Harder et al., 2021; 2022). Harder et al. (2022), for instance, pre-train a perceptual feature extractor using public data, then privatize the mean of the feature embeddings of the sensitive data records, and use the privatised mean to train a generative model.

**Limitations of previous work.** DP image generation based on custom training pipelines and architectures that are not used outside of the DP literature do not profit from the constant research progress on public image generation. Other works that instead build upon popular public generative approaches have been shown to not be differentially private despite such claims. This could be either due to faulty implementations or proofs. See Stadler et al. (2021) for successful privacy attacks on DP GANs, or Appendix B of Dockhorn et al. (2022) for an explanation on why DPGEN (Chen et al., 2022b) does not satisfy DP.

**Limited success on natural images.** DP synthesizers have found applications on tabular electronic healthcare records (Zhang et al., 2021; Torfi et al., 2022; Fang et al., 2022; Yan et al., 2022), mobility trajectories (Alatrista-Salas et al., 2022) and network traffic data (Fan & Pokkunuru, 2021). In the space of image generation, positive results have only been reported on MNIST, FashionMNIST and CelebA (downsampled to $32 \times 32$) (Harder et al., 2021; Wang et al., 2021; Liew et al., 2021; Bie et al., 2022). These datasets are relatively easy to learn thanks to plain backgrounds, class separability, and repetitive image features within and even across classes. Meanwhile, CIFAR-10 has been established as a considerably harder generation task than MNIST, FashionMNIST or CelebA (Radiuk, 2017). The images are not only higher dimensional than MNIST and FashionMNIST ($32 \times 32 \times 3$ compared to $28 \times 28$ feature dimensions), but the dataset has wider diversity and complexity. This is reflected by more complex features and textures, such as lightning conditions, object orientations, and complex backgrounds (Radiuk, 2017). Moreover, MNIST and Fashion-MNIST are considerably lower dimensional than CIFAR-10 and Camelyon ($28 \times 28$ vs $32 \times 32 \times 3$ features), and CelebA is downsampled to the same feature dimensionality as CIFAR-10 but has more than 3 times as many samples as CIFAR-10 (50k vs 162k) which considerably reduces the information loss introduced by DP training. As far as we know, only two other concurrent works have attempted DP image generation on

CIFAR-10. While Dockhorn et al. (2022) achieve a FID of only 97.7 by training a DP diffusion model from scratch, Harder et al. (2022) used pre-training on ImageNet and achieved the SOTA with a FID of 26.8, and a downstream accuracy of only 50%.

**Limited targeted evaluation.** The evaluation carried out on DP synthetic datasets is often not sufficiently targeted towards their utility in practice. The performance of DP image synthesizers is commonly evaluated on two types of metrics: 1) perceptual difference measures between the synthetic and real data distribution, such as FID, and 2) predictive performance of classifiers that are trained on a synthetic dataset of the size of the original training dataset and tested on the real test data. The former metric is known to be easy to manipulate with factors not related to the image quality, such as the number of samples, or the version number of the inception network (Kynkäänniemi et al., 2022). At the same time, it is not obvious how to jointly incorporate the information from both metrics given that they may individually imply different conclusions. Dockhorn et al. (2022), for instance, identify different diffusion model samplers to minimise either the FID or the downstream test loss. While some works have proposed evaluation frameworks for synthetic data outside of computer vision (e.g. Lin et al., 2020; Xu et al., 2019b; El Emam, 2020), it is not obvious how their analysis translates to the assessment of image generators. Further, recent research has identified use cases where synthetic data is not able to capture important first or second order statistics despite reportedly scoring highly on those metrics (Stadler et al., 2021; Stadler & Troncoso, 2022). In this paper, we set out to provide examples of additional downstream evaluations.

## 3 Background

### 3.1 Denoising Diffusion Probabilistic Models

Denoising diffusion models (Sohl-Dickstein et al., 2015; Song et al., 2020; Ho et al., 2020) are a class of likelihood-based generative models that recently established SOTA results across diverse computer vision problems (Dhariwal & Nichol, 2021; Rombach et al., 2022; Lugmayr et al., 2022). Given a forward Markov chain that sequentially perturbs a real data sample $x_0$ to obtain a pure noise distribution $x_T$, diffusion models parameterize the transition kernels of a backward chain by deep neural networks to denoise $x_T$ back to $x_0$.

Given $x_0 \sim q(x_0)$, one defines a forward process that generates gradually noisier samples $x_1, ..., x_T$ using a transition kernel $q(x_t|x_{t-1})$ typically chosen as a Gaussian perturbation. At inference time, $x_T$, an observation sampled from a noise distribution, is then gradually denoised $x_{T-1}, x_{T-2}, ...$ until the final sample $x_0$ is reached. Ho et al. (2020) parameterize this process by a function $\epsilon_\theta(x_t, t)$ (resp. $\epsilon_\theta(x_t, y, t)$ if $x_0$ has label $y$) which predicts the noise component $\epsilon$ of a noisy sample $x_t$ given timestep $t$. They then propose a simplified training objective to learn $\theta$, namely

$$\mathcal{L}(\theta|x_0) = \mathbb{E}_{t,x_t,\epsilon}[\|\epsilon - \epsilon_\theta(x_t, t)\|^2] = \mathbb{E}_{t,x_0,\epsilon}[\|\epsilon - \epsilon_\theta(\sqrt{\overline{\alpha}_t}x_0 + \sqrt{1-\overline{\alpha}_t}\epsilon, t)\|^2] \quad (1)$$

with $t \sim \mathcal{U}\{0, T\}$, where $T$ is the pre-specified maximum timestep, and $\mathcal{U}\{a, b\}$ is the discrete uniform distribution bounded by $a$ and $b$. The noisy sample $x_t$ is defined by $x_t = \sqrt{\overline{\alpha}_t}x_0 + \sqrt{1-\overline{\alpha}_t}\epsilon$ where $\epsilon \sim \mathcal{N}(0, I)$ is a noise sample of the same dimensionality as the data sample $x_0$, and $\overline{\alpha}_t$ is defined such that $x_t$ follows the pre-specified forward process. Most importantly, $\overline{\alpha}_t = \prod_{s=1}^{T}(1 - \beta_s)$ is a decreasing function of timestep $t$ where the form of $\beta_s$ is a hyperparameter. Thus, the larger $t$ is, the noisier $x_t$ will be.

### 3.2 Differential Privacy

Differential Privacy (DP) is a formal privacy notion that, in informal terms, bounds how much a single observation can change the output distribution of a randomised algorithm. More formally:

**Definition 3.1** (Differential Privacy (Dwork et al., 2006))**.** *Let $A$ be a randomized algorithm, and let $\varepsilon > 0$, $\delta \in [0, 1]$. We say that $A$ is $(\varepsilon, \delta)$-DP if for any two neighboring datasets $D, D'$ differing by a single element and $\mathcal{S}$ denoting the support of $A$, we have that*

$$\forall S \subset \mathcal{S}, \ \mathbb{P}[A(D) \in S] \leq \exp(\varepsilon)\mathbb{P}[A(D') \in S] + \delta.$$

The privacy guarantee is thus controlled by two parameters, $\varepsilon$ and $\delta$. While $\varepsilon$ bounds the log-likelihood ratio of any particular output that can be obtained when running the algorithm on two datasets differing in a single data point, $\delta$ is a small probability which bounds the occurrence of infrequent outputs that violate this bound (typically $1/n$, where $n$ is the number of training examples). The smaller these two parameters get, the higher is the privacy guarantee. We therefore refer to the tuple $(\varepsilon, \delta)$ as privacy budget.

### 3.3 Differentially Private Stochastic Gradient Descent

Neural networks are typically privatised with Differentially Private Stochastic Gradient Descent (DP-SGD) (Abadi et al., 2016), or alternatively a different DP optimizer like DP-Adam (McMahan et al., 2018). At each training iteration, the mini-batch gradient is clipped per example, and Gaussian noise is added to it. More formally, let $\mathcal{L}(\theta|x_0)$ denote the learning objective given model parameters $\theta \in \mathbb{R}^p$, input features $x_0 \in \mathbb{R}^d$ and label $y$. When training conditional image diffusion models, we estimate the loss function by $\widehat{\mathcal{L}}(\theta|x_0, y, t, \epsilon) = \|\epsilon - \epsilon_\theta(\sqrt{\overline{\alpha}_t}x_0 + \sqrt{1 - \overline{\alpha}_t}\epsilon, y, t)\|^2]$ where $x_0 \in \mathbb{R}^d, t \sim \mathcal{U}\{1, T\}, \epsilon \sim \mathcal{N}(0, I_d)$ and $I_p \in \mathbb{R}^{d \times d}$ being the identity matrix. Let $\texttt{clip}_C(v) : v \in \mathbb{R}^p \mapsto \min\left\{1, \frac{C}{\|v\|_2}\right\} \cdot v \in \mathbb{R}^p$ denote the clipping function which re-scales its input to have a maximal $\ell_2$ norm of $C$. For a minibatch $\mathcal{B} \subset D$ with $|\mathcal{B}| = B$ samples from the dataset $D$, the "privatised" minibatch gradient $\widehat{g}$ takes on the form

$$\widehat{g} = \frac{1}{B} \sum_{x_0 \in \mathcal{B}} \texttt{clip}_C\left(\nabla \mathcal{L}(\theta|x_0)\right) + \frac{\sigma C}{B}\xi,$$

with $\xi \sim \mathcal{N}(0, I_p)$. In practice, the choice of noise standard deviation $\sigma > 0$, batch-size $B$ and maximum number of training iterations are constrained by the predetermined privacy budget $(\varepsilon, \delta)$ which is tracked during training with a built-in privacy accountant (Abadi et al., 2016; Mironov et al., 2019). Crucially, the choice of hyperparameters can have a large impact on the accuracy of the resulting model, and overall DP-SGD makes it challenging to accurately train deep neural networks. On CIFAR-10 for example, the highest reported test accuracy for a DP model trained with $\epsilon = 8$ was 63.4% in 2021 (Yu et al., 2021). De et al. (2022) improved performance on image classification tasks and in particular obtained nearly 95% test accuracy for $\epsilon = 1$ on CIFAR-10, using 1) pre-training, 2) large batch sizes, and 3) augmentation multiplicity.

As part of this paper, we analyze to what extent these performance gains transfer from DP image classification to DP image generation by training diffusion models with DP-Adam. Diffusion models are inherently different model architectures that exhibit different training dynamics than standard classifiers which makes introduces additional difficulties in adapting DP training. First, diffusion models are significantly more computationally expensive to train. Indeed, they operate on higher dimensional representations than image classifiers, so that they can output full images instead of a single label. This makes each update step much more computationally expensive for diffusion models than for classification ones. In addition, diffusion models also need more epochs to converge in public settings compared to classifiers. For example, for a batch size of 128 samples Ho et al. (2020) train a diffusion model for 800k steps on CIFAR-10, while Zagoruyko & Komodakis (2016) train a Wide ResNet for classification in less than 80k steps. This high computational cost of training a diffusion model makes it difficult to finetune the hyperparameters, which is known to be both challenging and crucial for good performance (De et al., 2022). Second, and related to sample inefficiency, the noise inherent to the training of diffusion models introduces an additional variance that compounds with the one injected by DP-SGD, which makes training all the more challenging. Thus overall, it is currently not obvious how to efficiently and accurately train diffusion models with differential privacy.

## 4 Improvements towards Fine-Tuning Diffusion Models with Differential Privacy

Similarly to Dockhorn et al. (2022), we propose to train diffusion models with a differentially private optimizer. We will now elaborate on how to improve upon the naive application of DP-SGD.

**Recommendations from previous work.** De et al. (2022) identify pre-training, large batch sizes, and augmentation multiplicity as effective strategies in DP image classification. We adopted their recommendations in the training of DP diffusion models, and confirmed the effectiveness of their strategies to the task

of DP image generation. In contrast to the work of Dockhorn et al. (2022), where the batch-size only scales up to 2,048 samples, we implemented virtual batching which helps us to scale to up to 32,768 samples.

**Pre-training.** Pre-training is especially integral to generating realistic image datasets, even if there is a considerable distribution shift between the pre-training and fine-tuning distributions. We note that pre-training might not always be an option if 1) pre-trained models are not available, 2) it is too computationally expensive to pre-train the models, or 3) the privacy of the pre-training data set needs to be protected. However, it facilitates and is often necessary for obtaining results of acceptable quality on tasks other than MNIST. Multiple SOTA pre-trained models are further available online (Rombach et al., 2022; Dhariwal & Nichol, 2021) and can be used for commercial use-cases (Rombach et al., 2022). When starting training from a good pre-trained model, finetuning is known to be computationally less expensive than training from scratch.

Unless otherwise specified, we thus pre-train all of our models on ImageNet32 (Chrabaszcz et al., 2017). ImageNet has been a popular pre-training dataset used when little data is available (Raghu et al., 2019), to accelerate convergence times (Liu et al., 2020), or to increase robustness (Hendrycks et al., 2019).

**Augmentation multiplicity with timesteps.** De et al. (2022) observe that data augmentation, as it is commonly implemented in public training, has a detrimental effect on the accuracy in DP image classification. Instead they propose the use of augmentation multiplicity (Fort et al., 2021). In more detail, they augment each unique training observation within a batch, e.g. with horizontal flips or random crops, and average the gradients of the augmentations before clipping them. Similarly to Dockhorn et al. (2022), we extend augmentation multiplicity to also sample multiple timesteps $t$ for each mini-batch observation in the estimation of Equation 1, and average the corresponding gradients before clipping. In contrast to Dockhorn et al. (2022) where only timestep multiplicity is considered, we combine it with traditional augmentation methods, namely random crops and flipping. As a result, while Dockhorn et al. (2022) find that the FID plateaus for around 32 augmentations per image, we see increasing benefits the more augmentation samples are used (see Figure 9). For computational reasons, we limit augmentation multiplicity to 128 samples.

**Modified timestep sampling.** The training objective for diffusion models in Equation 1 samples the timestep $t$ uniformly from $\mathcal{U}\{0, T\}$ because the model must learn to de-noise images at every noise level. However, it is not straightforward that uniform sampling is the best strategy, especially in the DP setting where the number of model updates is limited by the privacy budget. In particular, in the fine-tuning scenario, a pre-trained model has already learned that at small timesteps the task is to remove small amounts of Gaussian noise from a natural-looking image, and at large timesteps the task is to project a completely noisy sample closer to the manifold of natural-looking images. The model behavior at small and large timesteps is thus more likely to transfer to different image distributions without further tuning. In contrast, for medium timesteps the model must be aware of the data distribution at hand in order to compose a natural-looking image. A similar observation has been recently made for membership inference attacks (Carlini et al., 2023; Hu & Pang, 2023): the adversary has been shown to more likely succeed in membership inference when it uses a diffusion model to denoise images with medium amounts of noise compared to high- or low-variance noised images. This motivates us to explore modifications of the training objective where the timestep sampling distribution is not uniform, and instead focuses on training the regimes that contribute more to modelling the key content of an image.

Motivated by this reasoning, we considered replacing the uniform timestep distribution with a mixture of uniform distributions $t \sim \sum_{i=1}^{K} w_i \cdot \mathcal{U}\{l_i, u_i\}$ where $\sum_{i=1}^{K} w_i = 1, 0 \leq l_0, u_K \leq T$ and $u_{k-1} \leq l_k$ for $k \in \{2, ..., K\}$. On CIFAR-10, we found the best performance for a distribution with probability mass focused within $\{30, ..., 600\}$ for $T = 1,000$ where timesteps outside this interval are assigned a lower probability than timesteps within this interval. We assume this is due to ImageNet-pre-trained diffusion models being able to denoise other (potentially unseen) natural images if only a small amount of noise is added. Training with privacy for small timesteps can then decrease the performance because more of the training budget is allocated to the timesteps that are harder to learn and because of the noisy optimization procedure. Even when training from scratch on MNIST, we observe that focusing the limited training capacity on the harder-to-learn moderate time steps increases test performance.

We combine all the strategies mentioned above and train DP diffusion models similarly to public diffusion models but replace the aggregated batch gradient with a privatised version. Please see algorithm 1 for an overview of the full training pipeline. We highlighted changes to public model training in blue. While the algorithm is defined as outputting the model parameters and the privacy budget $(\varepsilon, \delta)$, in practise we pre-specify the number of iterations, the batch size, the clipping norm and the privacy level, and optimise over the noise standard deviation such that the privacy accountant returns the desired privacy level $(\varepsilon, \delta)$.

---

**Algorithm 1:** Finetuning of conditional DP image diffusion models.

---

**Input:** Labelled dataset $D$, number of iterations $E$, maximum number of timesteps $T$, parameters of timestep distribution $\{w_i, l_i, u_i\}_{i=1}^K$, $\{\overline{\alpha}_t\}_{t=1}^T$, batch size $B$, clipping norm $C$, relative noise standard deviation $\sigma$, pre-trained denoising diffusion model $\epsilon_\theta(x_t, y, t)$ that given learned parameter vector $\theta$ and timestep $t$ maps the noised data $(x_t, y)$ to a denoised estimate $\widehat{x}_0 \sim \mathbb{R}^d$, optimization algorithm $\texttt{optim}(\theta, \widehat{g})$ as a function of the current parameter vector $\theta$, and the batch gradient $\widehat{g}$, augmentation method $\texttt{augment}(x_0)$ that returns a label-preserving augmentation of $x_0$, number of augmentation samples $A$

**Output:** DP diffusion model parameters $\theta$, privacy parameter $\varepsilon$

---

**for** $e = 1$ *to* $E$ **do**

    Sample batch $\mathcal{B} \subset D$ with $|\mathcal{B}| = B$;

    **for** $(x_0, y) \in \mathcal{B}$ **do**

        **for** $a = 1$ *to* $A$ **do**

            Sample augmented image $\widetilde{x}_{0,a} = \texttt{Augment}(x_0)$;

            Sample $t \sim \sum_{i=1}^K w_i \cdot \mathcal{U}\{l_i, \ldots, u_i\}$;

            Sample a noise vector $\epsilon \sim \mathcal{N}(0, I_p)$;

            Estimate the simplified loss function $\widehat{\mathcal{L}}(\theta | \widetilde{x}_{0,a}, y) = \|\epsilon - \epsilon_\theta(\sqrt{\overline{\alpha}_t} x_0 + \sqrt{1 - \overline{\alpha}_t}\epsilon, y, t)\|^2$;

            Compute the public gradient $\nabla_\theta \widehat{\mathcal{L}}(\theta | \widetilde{x}_{0,a}, y)$;

        Average the gradients of the augmented images $\widetilde{g}(x_0, y) = \frac{1}{A} \sum_{a=1}^A \nabla_\theta \widehat{\mathcal{L}}(\theta | \widetilde{x}_{0,a}, y)$

    Compute the privatised batch gradient $\widehat{g}(x_0, y) = \frac{1}{B} \sum_{(x_0, y) \in \mathcal{B}} \texttt{clip}_C(\widetilde{g}(x_0, y)) + \frac{\sigma C}{B}\xi, \xi \sim \mathcal{N}(0, I_p)$;

    Update model parameters $\theta \leftarrow \texttt{optim}(\theta, \widehat{g}_e)$;

**return** *DP model parameters* $\theta$ *and compute overall privacy cost $(\varepsilon, \delta)$ using a privacy accounting method (Abadi et al., 2016; Mironov et al., 2019)*;

---

# 5 Empirical Results on Differentially Private Image Generation and their Evaluation

## 5.1 Current Evaluation Framework for DP Image Generators

The FID (Heusel et al., 2017) has been the most widely used metric for assessing the similarity between artificially generated and real images (Dhariwal & Nichol, 2021), and has thus been widely applied in the DP image generation literature as the first point of comparison (Dockhorn et al., 2022). While the FID is the most popular metric, numerous other metrics have been proposed, including the Inception Score (Salimans et al., 2016), the Kernel Inception Distance (Bińkowski et al., 2018), and Precision and Recall (Sajjadi et al., 2018). For the calculation of these metrics, the synthetic and real images are typically fed through an inception network that has been trained for image classification on ImageNet, and a distance between the two data distributions is computed based on the feature embeddings of the final layer.

Even though these metrics have been designed to correlate with the visual quality of the images, they can be misleading since they highly vary with image quality unrelated factors such as the number of observations, or the version number of the inception network (Kynkäänniemi et al., 2022). They also reduce complex image data distributions to single values that might not capture what practitioners are interested in when dealing with DP synthetic data. Most importantly, they may not effectively capture the nuances in image quality the further apart the observed data distribution is from ImageNet. For example, CIFAR-10 images have to be upscaled significantly, to be fed through the inception network which will then capture undesirable artifacts

Table 1: A summary of the best results provided in this paper when training diffusion models with DP-SGD. We report the test accuracy of classifiers trained on different data sets. The highest reported current *SOTA* corresponds to classifiers trained on DP synthetic data, as reported in the literature. Here, [Do22] refers to Appendix F Rebuttal Discussions in Dockhorn et al. (2022), and [Ha22] to Harder et al. (2022). Our test accuracy (*Ours*) denotes the accuracy of a classifier trained on a synthetic dataset that was generated by a DP diffusion model and is of the same size as the original training data. Note that we also report the model size of our generative models (*Diffusion M. Size*). The *Non-synth* test accuracy corresponds to the test accuracy of a DP classifier trained on the real dataset, using the techniques introduced by De et al. (2022). *[De22] This number is taken from De et al. (2022) for $\epsilon = 8$.

| Dataset | Image Resolution | Diffusion M. Size | Pre-Training Data | Test Accuracy (%) | | | Privacy |
| | | | | SOTA | Ours | Non-synth | $(\epsilon, \delta)$ |
| --- | --- | --- | --- | --- | --- | --- | --- |
| MNIST | $28 \times 28$ | 4.2M | – | 98.1 [Do22] | 98.6 | 99.1 | $(10, 10^{-5})$ |
| CIFAR-10 | $32 \times 32 \times 3$ | 80.4M | ImageNet32 | 51.0 [Ha22] | 88.0 | 96.6 *[De22] | $(10, 10^{-5})$ |
| Camelyon17 | $32 \times 32 \times 3$ | 80.4M | ImageNet32 | - | 91.1 | 90.5 | $(10, 3 \cdot 10^{-6})$ |

introduced by upsampling. In contrast, MNIST images are digit-based and thus exhibit other variations than natural images, further diminishing the effectiveness of the evaluation.

An alternative way of comparing DP image generation models is by looking at the test performance of a downstream classifier trained on a synthetic dataset of the same size as the real dataset (Xie et al., 2018; Dockhorn et al., 2022), and tested on the real dataset. We argue that the way DP generative models are currently evaluated downstream, i.e. by evaluating a single model or metric on a limited data set, needs to be revisited. Instead, we propose to explore how synthetic data can be most effectively used for prediction model training and hyperparameter tuning.

This line of thinking motivates the proposal of an evaluation framework that focuses on *how* DP image generative models are used by practitioners. Similar reasoning has motivated work on developing a rigorous evaluation framework for synthetic data outside of the computer vision literature (Lin et al., 2020; Xu et al., 2019b; El Emam, 2020). As these works typically consider marginal distributions, correlative structures, or prediction tasks on tabular data, it is not obvious how to translate their suggestions to the image domain. As such, our experiments focus on two specific use cases for private synthetic image data: *downstream prediction tasks*, and *model selection*. *Downstream prediction tasks* include classification or regression models trained on synthetic data and evaluated on real samples. This corresponds to the setting where a data curator aims to build a production-ready model that achieves the highest possible performance at test time while preserving the privacy of the training samples. *Model selection* refers to a use case where the data curator shares the generative model with a third party that trains a series of models on the synthetic data with the goal to provide guidance on the model ranking when evaluated on the sensitive real data records. We hope that with these two experiments we cover the most important downstream tasks and set an example for future research on the development of DP generative models. After presenting our results within the evaluation framework that is commonly used in the current literature, we investigate the utility of the DP image diffusion model trained on CIFAR-10 for the aforementioned tasks in section 6.

## 5.2 Experimental Setup

We now evaluate the empirical efficiency of our proposed methods on three image datasets: MNIST, CIFAR-10 and Camelyon17. Please refer to Table 1 for an overview on our main results. For CIFAR-10, we provide additional experiments to prove the utility of the synthetic data for model selection in section 6. While these benchmarks may be considered small-scale by modern non-private machine learning standards, they remain an outstanding challenge for image generation with DP at this time.

We train diffusion models with a U-Net architecture as used by Ho et al. (2020) and Dhariwal & Nichol (2021). In contrast to Dockhorn et al. (2022), we found that classifier guidance led to a drop in performance. Unless otherwise specified, all diffusion models are trained with a privatized version of Adam (McMahan

et al., 2018; Kingma & Ba, 2014). The clipping norm is set to $10^{-3}$, since we observed that the gradient norm for diffusion models is usually small, especially in the fine-tuning regime. The privacy budget is set to $\epsilon = 10$, as commonly considered in the DP image generation literature. The same architecture is used for the diffusion model across all datasets. More specifically, the diffusion is performed over 1,000 timesteps with a linear noise schedule, and the convolutional architecture employs the following details: a variable width with 2 residual blocks per resolution, 192 base channels, 1 attention head with 16 channels per head, attention at the 16x16 resolution, channel multipliers of (1,2,2,2), and adaptive group normalization layers for injecting timestep and class embeddings into residual blocks as introduced by Dhariwal & Nichol (2021). When fine-tuning, the model is pre-trained on ImageNet32 (Chrabaszcz et al., 2017) for 200,000 iterations. All hyperparameters can be found in Table 3.

As a baseline, we also train DP classifiers directly on the sensitive data, using the training pipeline introduced by De et al. (2022), and additionally hyperparameter tuning the learning rate. Please refer to Table 5 for more details. It is not surprising that these results partly outperform the image generators as the training of the DP classifiers is targeted towards maximising downstream performance.

### 5.3 Training from Scratch ($\emptyset \rightarrow$ MNIST)

The MNIST dataset (LeCun, 1998) consists out of 60,000 $28 \times 28$ training images depicting the 10 digit classes in grayscale. Since it is the most commonly used dataset in the DP image generation literature, it is included here for the sake of completeness.

**Method.** We use a DP diffusion model of 4.2M parameters without any pre-training, with in particular 64 channels, and channel multipliers (1,2,2). The diffusion model is trained for 4,000 iterations at a constant learning-rate of $5 \cdot 10^{-4}$ at batch-size 4,096, with augmult set to 128, and a noise multiplier of 2.852. The timesteps are sampled uniformly within $\{0, \ldots, 200\}$ with probability 0.05, within $\{200, \ldots, 800\}$ with probability 0.9, and within $\{800, \ldots, 1,000\}$ with probability 0.05. To evaluate the quality of the images, we generate 50,000 samples from the diffusion model. Then we train a WRN-40-4 on these synthetic images (hyperparameters given in Table 4), and evaluate it on the MNIST test set.

**Results.** As reported in Table 1, this yields a SOTA top-1 accuracy of 98.6%, to be compared to the 98.1% accuracy obtained by Dockhorn et al. (2022). Crucially, we find that to obtain this SOTA result, it is important to bias the timestep sampling of the diffusion model at training time: this allows the model to get more training signal from the challenging phases of the generation process through diffusion. Without this biasing, we obtain a classification accuracy of only 98.2%.

### 5.4 Fine-tuning on a Medical Application (ImageNet32 $\rightarrow$ Camelyon17)

To show that fine-tuning works even in settings characterised by dataset shift from the pre-training distribution, we fine-tune a DP diffusion model on a medical dataset. Camelyon17 (Bandi et al., 2018; Koh et al., 2021) comprises 455,954 histopathological $96 \times 96$ image patches of lymph node tissue from five different hospitals. The label signifies whether at least one pixel in the center $32 \times 32$ pixels has been identified as a tumor cell. Camelyon17 is also part of the WILDS (Koh et al., 2021) leaderboard as a domain generalization task: The training dataset contains 302,436 images from three different hospitals whereas the validation and test set contain respectively 34,904 and 85,054 images from a fourth and fifth hospital. Since every hospital uses a different staining technique, it is easy to overfit to the hues of the training data with empirical risk minimisation. At the time of writing, the highest accuracy reported in the leaderboard of official submissions is 92.1% with a classifier that uses a special augmentation approach (Gao et al., 2022). The SOTA that does not fulfill the formal submission guidelines achieves up to 96.5% by pre-training on a large web image data set and finetuning only specific layers of the classification network (Kumar et al., 2022).

**Method.** First we pre-train an image diffusion model on ImageNet32, before finetuning it with $(10, 3 \cdot 10^{-6})$-DP on the training data, downsampled to $32 \times 32$, with a batch size of 16,384 for 200 steps. We tuned the hyperparameters to achieve the lowest FID on the training data, and used the out-of-distribution validation data to tune the downstream classifiers. The timestep is sampled with 0.015 probability from $\{0, \ldots, 30\}$,

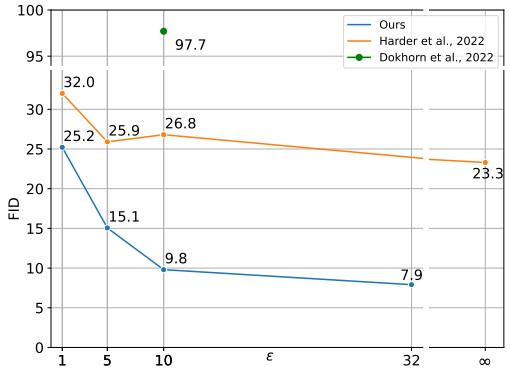
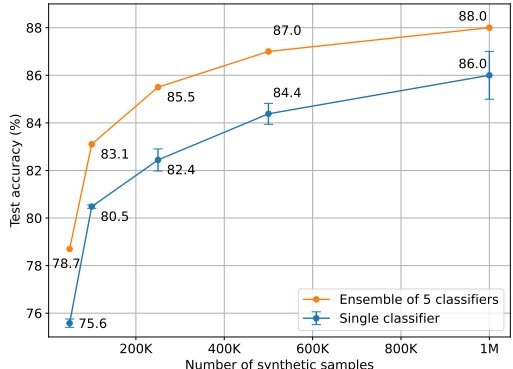

Figure 2: FID on CIFAR-10 for different privacy budgets. Our performance at $\epsilon = 5$ beats the pretrained SOTA for $\epsilon = \infty$. Results for Harder et al. (2022) are taken from their paper.

Figure 3: Downstream Top-1 accuracy of a CIFAR-10 WRN-40-4 as function of the number of synthetic data samples used to train it. The accuracy increases considerably with the dataset size.

with a probability of 0.785 in $\{30, \dots, 600\}$, and with 0.2 in $\{600, \dots, 1,000\}$. Since the diffusion model is pre-trained on ImageNet, we assume that the data is also available for pre-training the classifier. The pre-trained classifier is then further fine-tuned on a synthetic dataset of the same size as the original training dataset, which we find to systematically improve results. The classifiers were trained on augmented data where the augmentations include flipping, and color-jittering.

**Results.** We achieve close to SOTA classification performance with 91.1% by training only on the synthetic data whereas the best DP classifier we trained on the real dataset achieved only 90.5%. So while the synthetic dataset is not only useful for in-distribution classification, training on synthetic data is also an effective strategy to combat overfitting to domain artifacts and generalise in out-of-distribution classification tasks, as noted elsewhere in the literature (Zhou et al., 2020; Gheorghiță et al., 2022).

### 5.5 Fine-tuning on Natural Images (ImageNet32 → CIFAR-10)

CIFAR-10 (Krizhevsky et al., 2009) is a natural image dataset of 10 different classes with 50,000 RGB images of size $32 \times 32$ during training time.

**Method.** We use the same pre-trained model as for Camelyon17, that is an image diffusion model with more than 80M parameters trained on ImageNet32. We tune the remaining hyperparameters by splitting the training data into a set of 45,000 images for training, and 5,000 images for assessing the validation performance based on FID. As for Camelyon17, we found that sampling the timestep with probability 0.15 in $\{0, \dots, 30\}$, with 0.785 in $\{30, \dots 60\}$ and 0.2 in $\{600, \dots, 1,000\}$ gives us the lowest FID. More details can be found in Table 3.

**Results.** We improve the SOTA FID with ImageNet pre-training (Harder et al., 2022) from 26.8 to 9.8, which is a drop of more than 50%, and increase the downstream accuracy from 51.0% to 75.6% without pre-training the classifier. With pre-training the classifier on ImageNet32, we can achieve a classification accuracy of 86.6% with a single WRN. Modifying the timestep distribution led to a reduction in the FID from 11.6 to 9.8. Note that we achieved the results for different privacy levels by linearly scaling the number of iterations proportionally with $\epsilon$, and adjusting the noise level to the given privacy budget, keeping all parameters the same.

As detailed in Figure 2, we obtain SOTA accuracy for a variety of privacy levels. Even for a budget as small as $\epsilon = 1$, the FID obtained with our method is smaller than the current SOTA for $\epsilon = 10$. These results can also be compared with the very recent work by Dockhorn et al. (2022), who report an FID of 97.7 when training diffusion models with DP on CIFAR-10 without any pre-training. Due to the difficulty of the task

of learning the diffusion model with DP from scratch, the model did not learn to generate CIFAR-10-like samples, and the generated images do not seem to display any clear CIFAR-10 class instances at all. We believe that such mixed results are a clear motivation for our proposed method of pre-training on public data, which makes learning more tractable and realistic, and allows to obtain useful image generation.

To further confirm that the diffusion model has correctly learned the distribution shift, we trained a ResNet18 model to discriminate images from CIFAR-10 and ImageNet32 achieving a test accuracy of 98.0% on that task. We then evaluated it on 50,000 synthetically generated images out of which 92% were classified as CIFAR-10 images. This supports our hypothesis that the fine-tuned diffusion model does generate images that are more similar to CIFAR-10 than to the pre-training data of ImageNet.

### 5.6 Maximizing Downstream Prediction Performance by Sampling Arbitrary Many Data Records (ImageNet32 → CIFAR-10)

**Dataset sample size.** One benefit of synthetic data generators is their ability to render infinitely many synthetic images. As such, there is no reason why the comparison of real and synthetic samples should be limited to predictive models trained on the same number of training samples. We, therefore, investigate whether the performance of a downstream predictor increases with more training images. In Figure 3, we observe that the downstream classification accuracy constantly increases the more synthetic training observations are generated. In particular, we increase the downstream classification accuracy from 72.9% to 86.0% by sampling 1M instead of 50K images– without pretraining the classifier. We note that this difference is much more significant on the more challenging dataset of CIFAR-10 than e.g. MNIST, where we find that increasing the number of samples offers virtually no benefit in terms of downstream accuracy.

We note that the classifier can also be pre-trained on the pre-training distribution for performance increases for smaller data set sizes. The predictive performance does not increase significantly when fine-tuning on more samples (see the appendix - Figure 10). The benefit of pre-training the downstream classification performance thus diminishes for 1M synthetic samples.

**Ensembling.** We observe that we can further improve the classification accuracy given by a single WRN classifier by instead ensembling five different networks that differ only in the subsampling of the minibatches. As reported in Table 1, we can achieve a test accuracy of 88% on CIFAR-10 using this approach (see Figure 3).

## 6 Model Selection (ImageNet32 → CIFAR-10)

One important benefit of training a DP image generator over a DP classifier is the potential to use the generated data repeatedly for training a range of different prediction models and choosing the best one across them. Each experiment training a model on the data comes with a privacy cost, thus tuning a large number of DP classifiers increases the required privacy budget (Papernot & Steinke, 2021). In this section we consider how synthetic data can be used to gain initial insights on the choice of model, and to reduce the number of experiments run on the sensitive data records. The goal here is to identify the model that performs best on real data, while only having access to synthetic data. This becomes particularly relevant and useful when synthetic data needs to be released for research purposes (Chambon et al., 2022b), or data challenges (de Montjoye et al., 2014; Feuerverger et al., 2012; McFee et al., 2012).

For this purpose, we train 3 different model architectures that are commonly employed on CIFAR-10 (De et al., 2022; Bu et al., 2022a): a WRN-28-8 (Zagoruyko & Komodakis, 2016), a ResNet50 (He et al., 2016), and a VGG (Simonyan & Zisserman, 2014). For each architecture, we sweep over combinations of three to five different learning rates and three different values of weight decay. Please refer to Table 6 for more details. We now assess the utility of synthetic data for model selection in two stages of increasing difficulty. First, we check whether models – trained on the synthetic data – rank similarly on the real and synthetic test data. This corresponds to the application setting where a third party tunes a model on the synthetic data, and releases a model that is trained on the same data.

Once we have established that the test performance on real and synthetic data is sufficiently correlated to ensure that a model ranks similarly no matter which data set it was evaluated on, we train each model

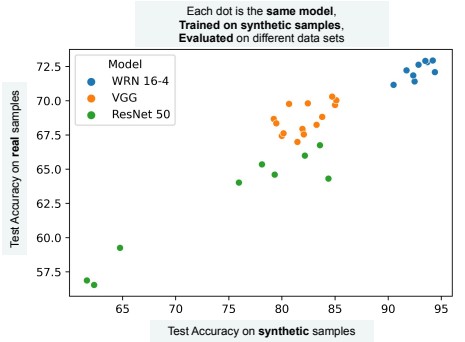

(a) Test accuracy on synthetic data vs real data of models trained on synthetic data

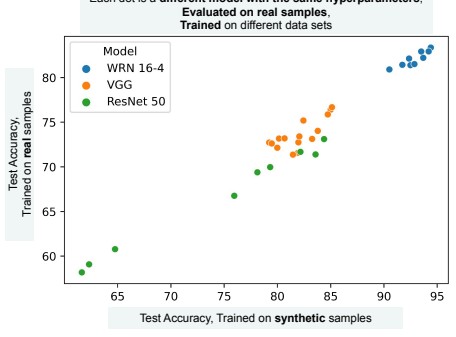

(b) Test accuracy of models trained and evaluated on synthetic data vs models trained and evaluated on real data

Figure 4: We observe that models rank similarly when evaluated on synthetic and real data. This suggests that findings on hyperparameter selection made on synthetic data can be transferred to the private dataset.

separately on 50K real and on the same number of synthetic samples. In both cases, models are tested on the same source of data they have been learned on (with sources being real or synthetic here). We then assess whether the test performance on the real and synthetic data is still correlated between models of the same architecture and hyperparameter constellation. This corresponds to the setting where a third party is responsible for finding the best model pipeline, but the data curator trains the final model with the optimal hyperparameters for their own internal use.

In Figure 4, we report our findings. We see that we can select the best or second best hyperparameter constellation in both application settings. More generally, we find that models rank similarly on both synthetic and real data, suggesting that findings with respect to relative model performance might transfer from the DP data to the original real data. However, we also notice that models overfit to the synthetic data distribution and that within one model group it is not obvious which hyperparameter constellation is the best. We therefore advice that synthetic data is used for a high-level orientation of the research direction. Note that the absolute test performance is not of interest here, only the correlation between the test performance on real and synthetic data as the best model will be released externally.

## 7 Conclusion

DP image generation has long attracted interest as a way of sharing synthetic data sets in sensitive application domains. Because of the degradation in performance introduced by DP-SGD, successful results on DP image generation have been limited to small and low-complexity data sets, like MNIST. In this paper, we set out to scale DP image generation to $32 \times 32 \times 3$ RGB image datasets. We proposed a methodology for DP diffusion models based on pre-training, timestep augmentation multiplicity, and a modified timestep sampling scheme. We are the first to train a DP image generator on a medical dataset where we achieved a downstream classification accuracy of 91.1% that is close to the SOTA of 96.5% with training on the real data. What is more, we also increased the SOTA downstream classification accuracy on CIFAR-10 from 51.0% to 88.0%. Recently proposed methods like latent diffusion models (Rombach et al., 2022) constitute a promising model class for DP fine-tuning on higher dimensional datasets, and we hope that our findings can contribute to future research exploring this direction.

Finally, we questioned how DP synthetic image data has been currently evaluated in the computer vision literature, and proposed an evaluation framework that is more suited to the needs of practitioners who would use the DP synthetic data as a replacement of the private dataset. For this purpose, we first considered maximising the downstream prediction performance by generating up to 1M data samples, and training ensembles. Second, we showed that findings from hyperparameter tuning on synthetic data translate to the corresponding findings on the real data.

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

## Supplementary Materials

## A    Comparison to Related Work

Please refer to Table 2 for a summary of details that differentiate our work from that of Dockhorn et al. (2022).

|  | **Dockhorn et al. (2022)** | **Ours** |
|---|---|---|
| Analysed for finetuning | False | True |
| Maximal parameter count | 1.8M | 80.4M |
| Time step scale | continuous | discrete |
| Classifier guidance | True | False |
| Modifications to time step sampling | False | True |
| Augmentation strategies | time step | time step, random crop, flipping |
| Number of samples for augmentation multiplicity | 32 | 128 |
| Maximal batch size | 2,048 | 16,384 |
| Evaluation | FID, downstream accuracy | FID, downstream accuracy, maximal downstream performance hyperparameter tuning |

Table 2: Comparison of our work and Dockhorn et al. (2022).

## B    Additional Experimental Details and Results

The code for the DP accounting and the classification models can be found on `https://github.com/deepmind/jax_privacy`. The implementation of the diffusion models follows `https://github.com/openai/guided-diffusion`. We present our hyperparameters in Tables 3, 4, 5 and 6.

**Details on evaluation metrics.**    To evaluate the generative performance of DP diffusion models, we estimate the FID and the downstream accuracy. For the former, we follow Dhariwal & Nichol (2021). For the latter, we generate as many samples from the diffusion model as the original private training data has, unless otherwise specified. We then train a classifier on the synthetic data set and report its test performance on the real test data. For the model selection experiments in section 6, we follow a similar procedure. Note that we here train and evaluate the downstream classifiers on the real or the synthetic data, as specified.

Example DP synthetic images can be found in Figures 5, 6 and 7. As a baseline, we also added samples from Dockhorn et al. (2022) when training from scratch on CIFAR-10 in Figure 8. This illustrates the importance of pre-training for large scale DP image generation. Please refer to Figure 9 for a sensitivity study of augmentation multiplicity. Finally, we show with class-wise metrics in Figure 11 that our model is not overfitting to a single class.

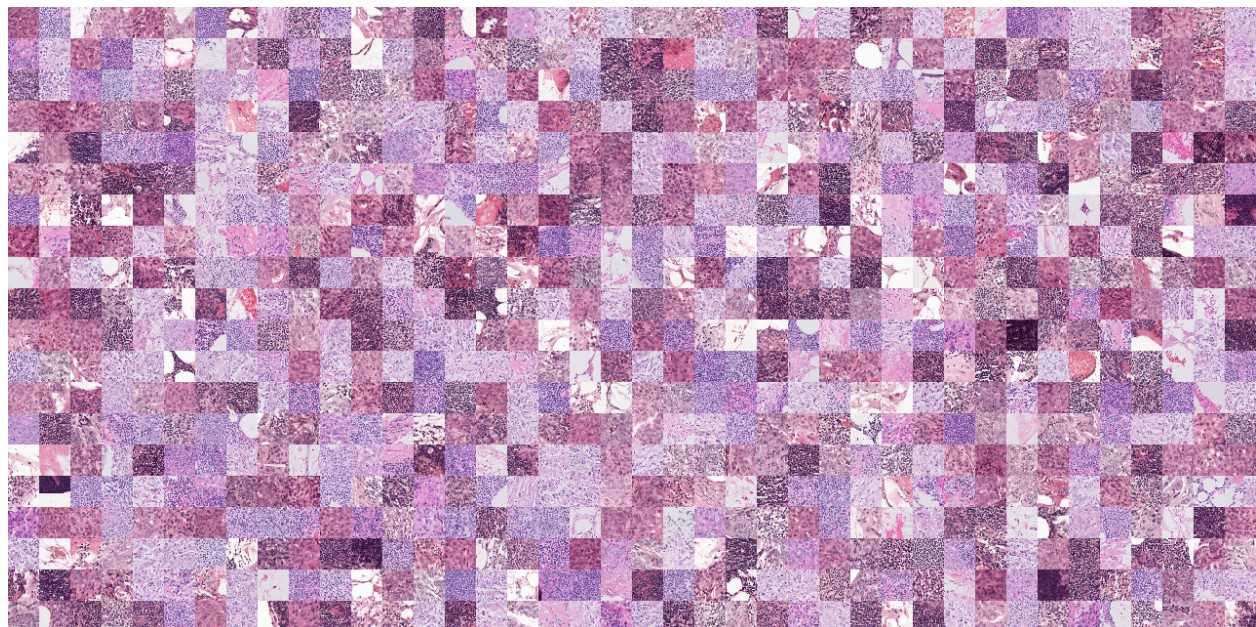

Figure 5: Random samples drawn from a DP image diffusion model trained on MNIST.

Figure 6: Random samples drawn from a DP image diffusion model trained on Camelyon17.

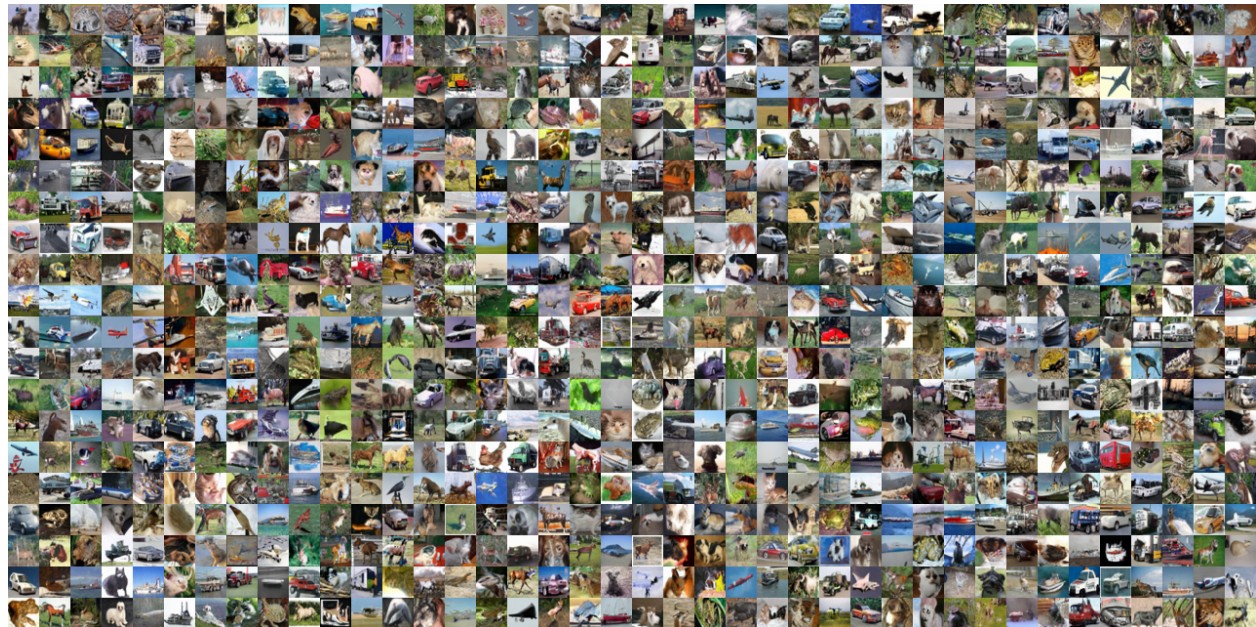

Figure 7: Random samples drawn from a DP image diffusion model trained on CIFAR-10.

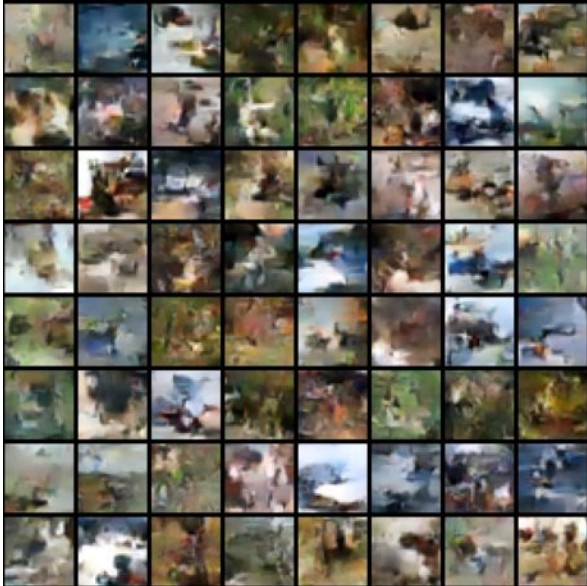

Figure 8: Samples from CIFAR-10 as provided by (Dockhorn et al., 2022) in Appendix F Rebuttal Discussions.

| | ImageNet32 | MNIST | CIFAR-10 | Camelyon17 |
|---|---|---|---|---|
| Pre-training data set | - | - | ImageNet32 | ImageNet32 |
| Privacy budget $(\epsilon, \delta)$ | $\infty$ | $(10, 10^{-5})$ | $(*varies, 10^{-5})$ | $(10, 10^{-5})$ |
| Iterations | 200k | 4,000 | 200 | 200 |
| Clipping norm | - | $10^{-3}$ | $10^{-3}$ | $10^{-3}$ |
| Noise schedule | linear | linear | linear | linear |
| Model size | 80.4M | 4.2M | 80.4M | 80.4M |
| Channels | 192 | 64 | 192 | 192 |
| Depth | 2 | 1 | 2 | 2 |
| Channels multiple | 1,2,2,2 | 1,2,2 | 1,2,2,2 | 1,2,2,2 |
| Heads channels | 64 | 64 | 64 | 64 |
| Attention resolution | 16 | 16 | 16 | 16 |
| Batch size | 1,024 | 4,096 | 16,384 | 16,384 |
| Learning rate | - | $5\times10^{-4}$ | $10^{-3}$ | $10^{-3}$ |
| Optimizer | Adam | Adam | Adam | Adam |
| Scheduler | linear$_{(from\ 0\ to\ LR\ in\ 5K\ steps)}$ | constant | constant | constant |
| $w_1, w_2, w_3$ | 0.03, 0.77, 0.2 | 0.05, 0.9, 0.05 | 0.015, 0.785, 0.2 | 0.015, 0.785, 0.2 |
| $l_1, u_1 = l_2, u_2 = l_3, u_3$ | 0, 30, 800, 1000 | 0, 200, 800, 1000 | 0, 30, 600, 1000 | 0, 30, 600, 1000 |
| # Augmentation samples | 0 | 128 (timestep) | 128 (timestep, flip) | 128 (timestep, flip) |
| Exponential moving average | 0.9999 | 0.9999 | 0.9999 | 0.9999 |

Table 3: Hyperparameters for diffusion models. The scale of the gradient noise is adjusted to ensure the desired privacy budget. *We report results for $\epsilon \in \{1, 5, 10, 32\}$. The implementation follows `https://github.com/openai/guided-diffusion`.

| | MNIST | CIFAR-10 | Camelyon17 |
|---|---|---|---|
| Pre-training data set | - | - | ImageNet32 |
| Iterations | 10,000 | 20,000 | 4,000 |
| Depth | 40 | 40 | 40 |
| Width | 4 | 4 | 4 |
| Dropout | 0.5 | 0.0 | 0.0 |
| Weight decay | $5 \cdot 10^{-4}$ | $5 \cdot 10^{-4}$ | $1 \cdot 10^{-5}$ |
| Label smoothing | 0.05 | 0.0 | 0.0 |
| Batch size | 64 | 4,096 | 512 |
| Learning rate | 0.1 | cosine decay$_{(start\ at\ 0.1\ with\ \alpha = 0,\ 1\ decay\ step)}$ | $10^{-4}$ |
| Optimizer | SGD | SGD | SGD |
| Nesterov's momentum | 0.9 | 0.9 | 0.9 |
| Samples augmentation multiplicity | 0 | 0 | 16 |
| Exponential moving average | 0.9999 | 0.9999 | 0.9999 |
| # Augmentation samples | 1 (crop) | 16 (crop, flip, color) | 16 (color, flip, crop) |

Table 4: Hyperparameters for downstream classification WRNs trained on synthetic data.

|                                    | MNIST          | Camelyon17            |
| ---------------------------------- | -------------- | --------------------- |
| Privacy budget $(\epsilon, \delta)$ | $(10, 10^{-5})$ | $(10, 10^{-5})$      |
| Pre-training data set              | -              | ImageNet32            |
| Clipping norm                      | 1              | 1                     |
| Noise standard deviation           | 3.0            | 4.0                   |
| Depth                              | 16             | 40                    |
| Width                              | 4              | 4                     |
| Batch size                         | 16,384         | 4,096                 |
| Learning rate                      | 4.0            | 0.5                   |
| Optimizer                          | SGD            | SGD                   |
| Samples augmentation multiplicity  | 0              | 16 (color, flip, crop) |
| Exponential moving average         | 0.9999         | 0.9999                |

Table 5: Hyperparameters for DP WRN classifiers trained on the sensitive data records. Training was stopped once $\epsilon = 10$ was reached.

|                                   | ResNet50                                      | VGG                                        | WRN-16-4           |
| --------------------------------- | --------------------------------------------- | ------------------------------------------ | ------------------ |
| Learning rate                     | $5 \cdot 10^{-4}, 2 \cdot 10^{-3}, 4 \cdot 10^{-3}$ | $0.07, 0.04, 0.025, 0.01, 5 \cdot 10^{-3}$ | $0.01, 0.02, 0.03$ |
| Weight decay                      | $0, 0.1, 0.01$                                | $0.0, 10^{-3}, 5 \cdot 10^{-3}$            | $0.0, 0.1, 0.01$   |
| Iterations                        | 15,000                                        | 15,000                                     | 15,000             |
| Label smoothing                   | 0.05                                          | 0.0                                        | 0.0                |
| Batch size                        | 128                                           | 128                                        | 128                |
| Optimizer                         | SGD                                           | SGD                                        | SGD                |
| Momentum                          | 0.0                                           | 0.9                                        | 0.9                |
| Scheduler                         | cosine decay*                                 | constant                                   | cosine decay*      |
| Samples augmentation multiplicity | 16 (flip, crop)                               | 0                                          | 16 (flip, crop)    |
| Exponential moving average        | 0.9999                                        | 0.9999                                     | 0.9999             |

Table 6: Hyperparameters for classifiers for the model selection experiment. All combinations of different values displayed for learning rate and weight decay were trained. *Cosine decay schedule with initial learning rate swept over, 1 decay step, and $\alpha = 0.0$.

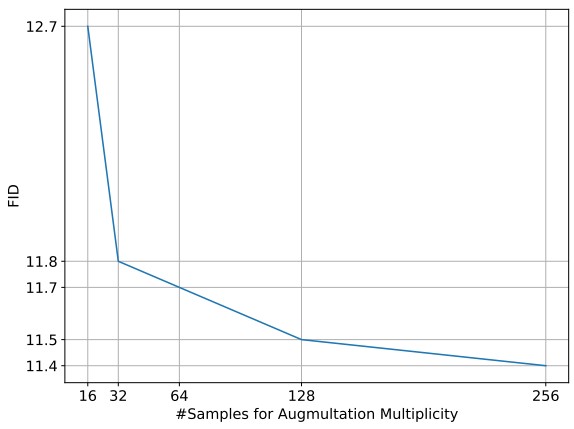

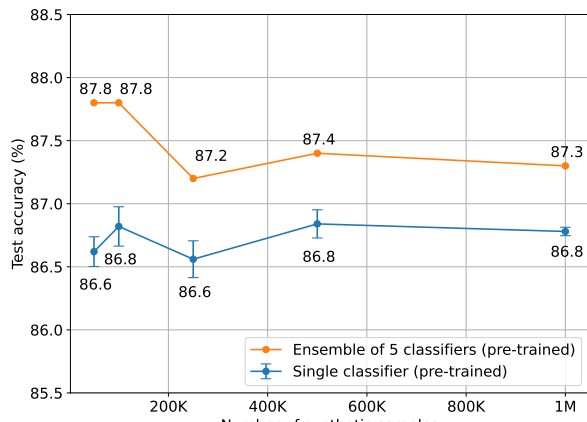

Figure 9: FID on CIFAR-10 for different values of augmentation multiplicity samples. We see that the FID is decreasing for values up to 256.

Figure 10: Downstream accuracy of an ImageNet32 pre-trained CIFAR-10 WRN-40-4 as function of the number of synthetic data samples used to train it.

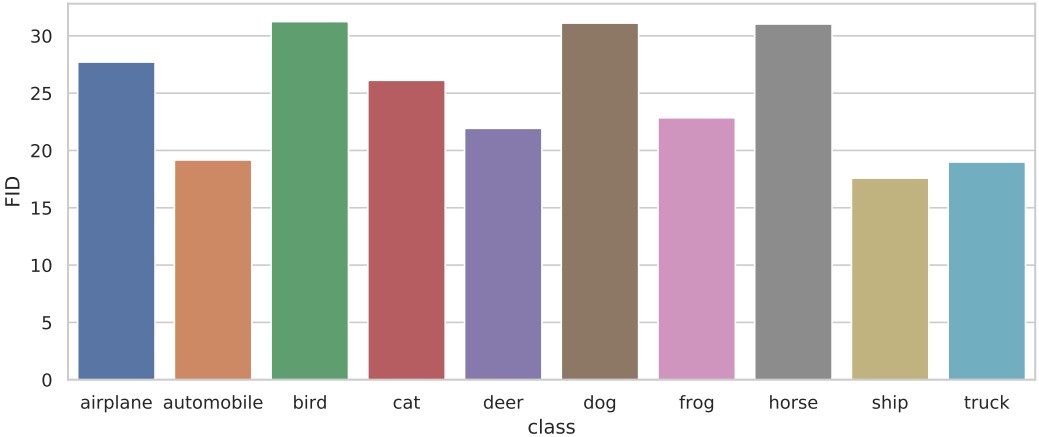

(a) Percentage of samples per class classified as CIFAR-10 images by ResNet18 model trained to discriminate CIFAR-10 and ImageNet32 images.

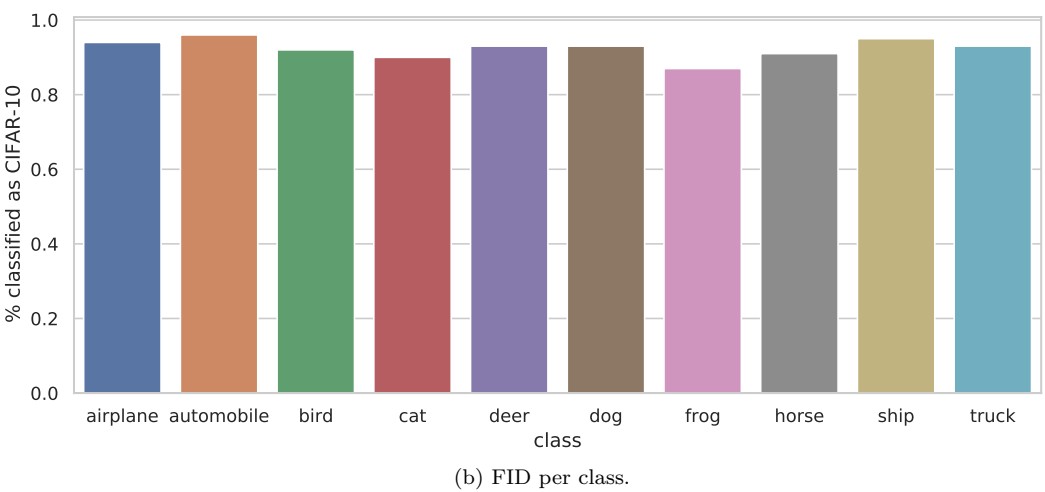

(b) FID per class.

Figure 11: Class-wise metrics on CIFAR-10.

