# OpenReview forum: "Differentially Private Diffusion Models Generate Useful Synthetic Images"
_TMLR — Rejected by TMLR_

### Review · Reviewer_hDPw · 2023-04-08

**Summary Of Contributions:**

The paper shows that by carefully choosing the hyper-parameters, differentially private (DP) diffusion models can generate images that are both high-quality (in terms of FID) and useful for downstream applications including classification and model selection. More specifically, the paper establishes new state-of-the-art scores on a series of benchmark datasets, including FID on CIFAR10 and downstream classification accuracy on Camelyon17.

**Audience:**

Yes

**Broader Impact Concerns:**

No concerns.

**Claims And Evidence:**

Yes

**Requested Changes:**


Overall, the paper is of good quality and is a good fit for TMLR standards. I vote for acceptance if the following issues can be addressed in the rebuttal and the revision.

* Section 5.4: it says that the data augmentations used include "flipping, rotation and color-jittering". However, in Table 4 in the appendix, "rotation" is not listed.

* In Sections 5.5 and 5.6, it says that the downstream classification accuracy on CIFAR10 trained with 50,000 generated samples is 72.9%. However, according to Figure 3, this number should be 75.6%.

* Section 5.1 says "This line of thinking motivates the proposal of an evaluation framework that focuses on how DP generative models are used by practitioners." Section 7 says "Finally, we questioned how DP synthetic data has been currently evaluated in the literature, and proposed an evaluation framework that is more suited to the needs of practitioners who would use the DP synthetic data as a replacement of the private dataset." I like the idea and I agree that this evaluation strategy is more informative. However, I want to point out that similar evaluation approaches (with the same motivation) have already been proposed in prior work back in 2019 (e.g., see Section 2.1 and Section 5.2 in [1])

* The downstream classification accuracy on CIFAR10 with a pre-trained classifier (86.6%) is mentioned both in Section 5.5 and Section 5.6. It is more concise to keep one of them.

* Page 9: "[600, 1000].Since" -> "[600, 1000]. Since"

[1] Lin, Zinan, et al. "Using gans for sharing networked time series data: Challenges, initial promise, and open questions." Proceedings of the ACM Internet Measurement Conference. 2020.

**Strengths And Weaknesses:**


Strengths:
* The paper establishes new state-of-the-art results on DP synthetic image generation.
* The insights and experience in tuning DP diffusion models (Section 4) can be useful for future work in the domain.
* Overall, the paper is well-written.

Weaknesses:
* There are some potential typos in the paper. See "Requested Changes" section.

---

### Review · Reviewer_464t · 2023-04-10

**Summary Of Contributions:**

The paper discusses the generation of privacy-preserving synthetic images using differentially private diffusion models. The authors propose a simple methodology, including timestep-dependent augmentation and timestep sampling of the diffusion models that allows them to generate high-quality synthetic image datasets that are useful for various downstream tasks. They show that by privately fine-tuning ImageNet pre-trained diffusion models, they can achieve better results on CIFAR-10 and Camelyon17 in terms of both FID and the accuracy of the downstream classifiers.

**Audience:**

Yes

**Broader Impact Concerns:**

I did not find significant broader impact concerns of this paper.

**Claims And Evidence:**

No

**Requested Changes:**

> Some references need to be completely cited. For example

- Rezende, D. J., S. Mohamed, and D. Wierstra (2014). “Stochastic backpropagation and approximate inference in deep generative models”. In: International Conference on Machine Learning. 1278–1286.

- Song, Y., Sohl-Dickstein, J., Kingma, D. P., Kumar, A., Ermon, S., & Poole, B. Score-Based Generative Modeling through Stochastic Differential Equations. In International Conference on Learning Representations.

are respectively important references for variational autoencoders and diffusion models, but not cited. The authors might need a more careful check for the completeness of the reference.


> The math expressions need to be polished:

Eq (1): the expectation is wrt $x_0$ but did not involve $x_t$.

In 3.3, i (the data index) is not explained in the text.

In the paper, the authors mainly discuss time-discrete diffusion models such as DDPM, but use [0,T], which refers to a continuous interval, to present the timestep range.

Also, in the paragraph above section 5, it is not clear $[l_i, u_i]$ indicating a discrete or continuous time interval.

After reading the comparison in the appendix Table 2, I find the authors apply continuous time steps. This part is not clearly explained in the method part. The authors may need more clarification regarding the diffusion schedule and how to discretize in generation.

> The presentation of section 4

After reading through section 4, it is yet not clear how the authors fine-tune the diffusion models with differential privacy. The authors suggests a series of improvement in terms of timestep-dependent augmentation, the sampling of timesteps, etc, but assume the readers have strong knowledge to De et al (2022) and Dockhorn et al (2022). It would be clear if the authors could give a short introduction about how to train DP diffusion models (e.g., what's the loss function).


> Experimental sections

The authors discuss the efficiency of both training and sampling diffusion models with DP. However, the experiments do not contain the results indicating the proposed method is more efficient in the training. The authors may need to  consider to either narrow down the scope of discussion or add corresponding experiments to support the claim.

> The comparison with baselines

In figure 2, the authors compare FID with Harder et al, 2022, while diffusion models are not applied in such a baseline. This comparison may not be fair. A more straightforward baseline could be Dockhorn et al. (2022).

The authors needs to further explain the most important difference other than the augmentation and timestep sampling, when compared with Dockhorn et al. (2022).

In Dockhorn et al. (2022)., it seems the diffusion model is time-continuous, while in Table 2, they are classified as a discrete timestep scale. This part needs further clarification.

**Strengths And Weaknesses:**

### Strengths

The paper is overall well-written and organized, making it easy to follow. The proposed method is well-motivated and has been shown to be effective. The authors demonstrate the improvements in terms of both the downstream classifier and the generation. The experimental details, such as implementation, experiment settings are provided, and should be sufficient for reproducibility.

### Weakness

The motivation for training diffusion models with differential privacy is a bit unclear. Although it is interesting to see this exploration, the paper does not provide sufficient theoretical support nor strong empirical evidence (compared to normal diffusion models).

The proposed method shows improvements in terms of FID compared to Harder et al, 2022, which does not apply diffusion model. Such improvement is not sufficient to demonstrate the advantages of training/fine-tuning diffusion models with DP. Comparing with the baselines in terms of classification performance, the improvements are a bit marginal.

The proposed method is claimed not to be computationally expensive. However, no experiment is included in the paper to support this perspective.

The paper does not present the method clearly and some comparison is confusing; please see the requested changes for more details.

---

### Review · Reviewer_N6L9 · 2023-04-20

**Summary Of Contributions:**

This paper studies the problem of generating useful and privacy-preserving synthetic data based on diffusion models. Also, the paper considers the pretraining and finetuning pipeline. Built upon DP-SGD, the paper proposes a method to privately fine-tune pre-trained ImageNet diffusion models on local (sensitive) data, achieving better empirical performance.



**Audience:**

Yes

**Broader Impact Concerns:**

There are no broader impact concerns for this paper.

**Claims And Evidence:**

Yes

**Requested Changes:**

1. I request the authors clarify how your proposed method is positioned with respect to related works on privacy-preserving image generation. The clarification should explain the advantages and limitations of using pretraining data/models.

2. I request the authors clarify the proposed method's technical contributions compared with existing works. Discussing the technical challenges and explaining the reasoning behind the improved empirical performance is suggested.

3. A clear description of the proposed method and additional evaluation metrics should be provided, so readers can easily understand how the method and evaluation are implemented.

**Strengths And Weaknesses:**

Strengths:

1. The paper studies privacy-preserving image generation based on diffusion models, which is a timely problem, given that diffusion models have gained much attention recently.

2. The proposed method shows good empirical performances in terms of both FID scores and downstream classifications, which is promising.

3. Additional evaluation metrics are proposed for practical considerations.

Weaknesses:

1. The setting/motivation for using pretraining data sets and/or pretrained diffusion models is not well-explained. This is a different setting compared with most existing literature on privacy-preserving image synthesis, such as [1]. I have two concerns related to this: (1) Since pretrained diffusion models are usually very large, the party wanting to share the sensitive data must have enough computational resources to fine-tune the model. So is it practical? (2) The comparisons of your method to [1] and other methods that do not use the information of pretraining data are not fair.

2. The novelty of the proposed method seems limited, given existing works [1,2]. All the elements introduced in Section 4 are not new. The improved performance demonstrated in the experimental sections seems to come from using the pretrained diffusion model. It is unclear what the technical challenges are when privately fine-tuning the pretrained diffusion models using DP-SGD.

3. The proposed method and the additional evaluation metrics are not well-explained. I have to read the related works to understand how the proposed method is implemented. I recommend providing a pseudocode of the overall training algorithm and a clear definition of the proposed additional evaluation metrics.

[1] Tim Dockhorn, Tianshi Cao, Arash Vahdat, and Karsten Kreis. Differentially private diffusion models. arXiv preprint arXiv:2210.09929, 2022.

[2] Soham De, Leonard Berrada, Jamie Hayes, Samuel L Smith, and Borja Balle. Unlocking high-accuracy differentially private image classification through scale. arXiv preprint arXiv:2204.13650, 2022.

---

### Decision · Action_Editors · 2023-05-20

**Recommendation:** Reject

**Comment:**

The reviewers raised a number of questions during the discussion. After the discussion, two reviewers' concerns have been cleared out, and one remains unconvinced. The main concern being that it is hard to see if there are any new implications for privacy-preserving image synthesis besides using a pre-trained model, and the comparisons is not fair as the baselines did not use pretrained models, so the performance gain did not really comes from the DP side. I agree with this, and I think more extensive evaluations should be performed to demonstrate the claim.

Due to several limitations, I have to recommend rejection this time, but I encourage the authors to revise the paper as suggested by the reviewers and my comments below, and submit a new version for another round of reviewing, which I am happy to monitor it if necessary.

1. The paper provides several tricks to demonstrate the feasibility and usefulness of DP training a diffusion models. I am not criticizing on the technical contributions, but I think the proposed tricks need more careful investigations, for example, how much gain can each trick get in the DP setting? I believe evaluations with individual tricks applied need to be performed.

2. Some presented results are confusing. For example, Table 1 is claimed to be evaluated with a model trained from scratch in Section 5.3?. Is this for all the datasets? or just MNIST? If it is for all datasets, the results in Section 5.5 are confusing as they are much worse than what are reported in Table 1. If only MNIST is trained from scratch, Table 1 is really confusing as it mixes results from different settings. In addition, I think both results trained from scratch and from pretrained models should be reported and compared.

3. Since this paper is mainly about empirical work, I think the current evaluations on the 3 datasets are not sufficient. There have been a lot of downstream datasets created these years to evaluate the performance of a model. I encourage the authors to evaluate on more datasets, e.g., some datasets from the benchmark: https://computer-vision-in-the-wild.github.io/ELEVATER/

**Audience:**

Audience from DP and diffusion model community can find the work useful.

**Claims And Evidence:**

This paper claims that diffusion models fine-tuned with differential privacy can produce useful and provably private synthetic data, and performs several experiments to demonstrate the performance in the setting. Although the provided results somehow support the claims, I found it is not sufficient and convincing enough. Please see the comments below.